# Stimulatory Effect of Lactobacillus Metabolites on Colonic Contractions in Newborn Rats

**DOI:** 10.3390/ijms24010662

**Published:** 2022-12-30

**Authors:** Constantin V. Sobol

**Affiliations:** Sechenov Institute of Evolutionary Physiology and Biochemistry, Russian Academy of Sciences, Thorez pr. 44, 194223 St. Petersburg, Russia; peep9@yandex.ru

**Keywords:** colon contraction, gut smooth muscle, microbiota, fermented/hydrolyzed products, metabolites, gastrointestinal motility, constipation, intracellular calcium

## Abstract

Microbiota are known to play an important role in gastrointestinal physiology and pathophysiology. Microbiota and their metabolites can affect gut motility, neural regulation and the enteric endocrine systems and immune systems of the gut. The use of fermented/hydrolyzed products may be a promising new avenue for stimulating gastrointestinal motility. The purpose of this study was to investigate the effect of lactobacillus metabolites (PP), produced using a U.S.-patented fermentation method, on rat colon motility in vitro. The distal colon was incised from newborn male Wistar rats. A sensitive tensometric method for the study of colon contractions was used. The [Ca^2+^]_i_ in colon tissue was registered using a computerized ratiometric system for an intracellular ion content assay (Intracellular Imaging and Photometry System, Intracellular imaging, Inc. Cincinnati, OH, USA). The cumulative addition of PP induced contraction with sigmoid dose responses with ED_50_ = 0.13 ± 0.02% (*n* = 4), where 10% PP was accepted as a maximal dose. This contraction was accompanied by an increase in the concentration of [Ca^2+^]_i_. It was shown that introducing Lactobacillus metabolites produced using a U.S.-patented fermentation method quickly stimulates dose-dependent colon contractions and an increase in intracellular calcium. The direct application of PP via enema to the colon could stimulate colon motility and suppress pathogenic microbiota, owing to the antagonistic property of PP on pathogens.

## 1. Introduction

Microbiota are known to play an important role in gastrointestinal (GI) physiology and pathophysiology. Microbiota and their metabolites can affect gut motility, neural regulation and the enteric endocrine and immune systems of the gut [1,2,3,4,5,6]. Microbiota produce many active metabolites/compounds, such as short-chain (SCFA) and long-chain fatty acids, vitamins, neurotransmitters, hormones, bile acids and others, which are involved in neuroendocrine communication within the host [3,6,7,8,9]. The products of microbiota fermentation can be involved in both physiological and pathophysiological processes in the host [5,6,8,9]. Interactions between microbiota-derived metabolites and host signaling molecules may be considered “host–microbial endocrinology” [8]. Some metabolites of microbiota, such as SCFAs, can be involved in GI motility and can modulate general motility and peristalsis [1,4]. One particularity of orally introducing additional microbial metabolites of symbiotic microorganisms is that they appear to modulate GI contractility without negatively affecting the host microbiota [10]. Microbiota and their metabolites can be changed in constipated patients [10,11,12] and in persons with neurodegenerative diseases [13,14]. GI dysfunction can significantly precede the onset of motor symptoms in Parkinson’s disease (PD) [13]. There is age-related impairment in GI function, together with reduced intestinal motility [15]. Many elderly people suffer from constipation [10,16]. At the cellular level in elderly rats, disruption has been reported in the intracellular signal transduction cascade in colon smooth muscle cells [17,18,19], and a reduction in agonist-induced contraction in GI smooth muscle cells has also been noted [18,19,20]. To improve GI function and increase GI motility, beneficial microorganisms and/or their metabolites can be utilized [9,10,21,22,23,24,25,26]. The purpose of this study was to investigate the effect of lactobacillus metabolites produced using a U.S.-patented fermentation method [27] (probiotic product, PP) on rat colon motility in vitro.

## 2. Results and Discussion

In the colon of a newborn rat, PP induced dose-dependent contractions (Figure 1). A dose–response curve links the concentration of a substance in the organ bath with the intensity of its effect on the preparation (Figure 1B). The dose that elicits 50% of the maximum possible effect is ED_50_ (median effective dose). In our experiments, the maximal dose was restricted to 10% of PP. ED_50_ in our experiments was equal to 0.13 ± 0.02% (*n* = 4) (Figure 1B).

This contraction was accompanied by an increase in the concentration of intracellular free Ca^2+^, [Ca^2+^]_i_ (Figure 2). The Ach-induced Ca^2+^ rise is shown for comparison. At a 10% PP concentration, a huge [Ca^2+^]_i_ elevation was observed in the colon segments. Calculation of the dose effect for [Ca^2+^]_i_ was not performed. It should be noted that contractions in the colon occurred rather quickly and may be considered phasic contractions. PP contains approximately 90 mM lactate, so 10% PP contains less than 10 mM. The addition of 10 mM lactate into the bath did not induce any rat colon contractions.

One of the main functions of the GI tract is motor activity. Smooth muscle cells, along with regulation by the autonomic and enteral nervous system, determine peristalsis. Deterioration in GI motility is associated with significant morbidity and increased health care costs [28]. GI motility also decreases with age [15,16]. Many elderly people suffer from constipation [10,16]. At the cellular level, reduced motility has reportedly been associated with the disruption of the intracellular signal transduction cascade in colon smooth muscle cells [17,18] and a reduction in agonist-induced contractions in GI smooth muscle cells in elderly rats [18,19,20]. Motility disorders (both slowing down and speeding up) lead to disruption of the normal processes of digestion and absorption and therefore to a change in the composition of the internal environment of the gut, which, in turn, affects the composition of the host gut microbiota, potentially stimulating the development of dysbacteriosis.

Microbiota play a significant role in the development and maintenance of normal gut physiology and motility [4,29]. By virtue of the mix of metabolites produced, microbiota is capable of modulating GI motility directly via smooth muscle cells and indirectly via the enteric nervous system, the immune system and neuroendocrine systems in the gut [4,28,29,30].

Probiotics and/or their metabolites have been used for the safe modulation of GI motility and improvement in gut physiology [4,10,11,21,22,23,24,25,31]. In eligible randomized controlled trials, it was found that the consumption of multispecies probiotics might improve GI motility and stool consistency [26]. Long-term consumption of a multistrain probiotic supplement was effective and safe in the elderly with functional constipation [10]. Colonization of germ-free mice with *Lactobacillus acidophilus* together with *Bifidobacterium bifidum* restored GI motility to some extent [30].

The use of fermented/hydrolyzed products may be a promising avenue for stimulating GI motility. These products contain various amino acids, active peptides, enzymes, vitamins and other beneficial ingredients [27,32], which can readily be utilized in the gut. The administration of fermented milk prepared with *Lactobacillus casei* strain Shirota increased colonic activity in pigs [25]. Fermented milk with *Lactobacillus casei* may alleviate constipation-related symptoms in women [21]. Whey protein hydrolysate was shown to increase rat colon motility in an in vitro model [23]. Cell-free supernatants of *Escherichia coli* Nissle 1917 increased the maximal tension forces of smooth muscle strips from a human colon in vitro model [24]. Some authors considered secreted metabolites of beneficial bacteria to potentially be an alternative microecological therapy for constipation in patients with impaired host defenses [22].

This study tested the effects of a proprietary probiotic/pharmabiotic product, the result of the near-complete fermentation of food products with specially selected lactobacilli [27]. The goal of this U.S.-patented fermentation process is to fully digest select food ingredients, achieving maximal beneficial bacterial metabolite concentrations [27]. This fermentation process imitates human gut digestion, but it lasts for more than three days. The end product can easily be adsorbed by the GI tract without additional digestion. PP suppresses the growth of pathogenic microbiota because of its high acidity (pH < 3.0) and the presence of a high concentration of metabolites of beneficial bacteria [27]. PP inhibited the growth of gut-associated pathogens, such as *Vibrio cholerae*, *Salmonella enterica* typhi and *Shigella*
*sonnei*, and stimulated the growth and reproduction of the host’s beneficial microbiota [27]. In other studies, it was demonstrated that cell-free supernatants produced by Lactobacillus strains revealed antibacterial activity in vitro [33]. Thus, metabolites of lactic acid bacteria may contain antimicrobial ingredients.

In this study, it was shown that, in the colons of newborn rats, PP induced dose-dependent contractions. These contractions were accompanied by an increase in [Ca^2+^]_i_. At high concentrations of PP, a huge [Ca^2+^]_i_ elevation and contraction were observed in colon segments. It is known that [Ca^2+^]_i_ initiates and supports smooth muscle contraction [34]. Earlier, it was shown that PP strongly stimulates Ca^2+^ release from intracellular stores in various cell types, such as PC-12, rat brain neurons and aortic smooth muscle [6,27]. The fact that contraction in the colon occurs rather quickly suggests that PP may also release Ca^2+^ from intracellular stores in colon smooth muscle cells. These effects will be evaluated in subsequent experiments. They are important to evaluate because, in elderly rats, Ca^2+^ mobilization from IP_3_-sensitive stores was shown to be impaired [19], and PP could stimulate Ca^2+^ release after local application.

PP contains approximately 90 mM lactate, so 10% PP is less than 10 mM. The addition of 10 mM lactate into the bath did not induce any rat colon contractions. Lactobacilli were absent in PP in these in vitro experiments. Thus, neither lactobacilli nor lactate were responsible for receptor activation in the colon segments used in these experiments. Other unidentified Lactobacillus metabolites are most likely responsible for colon contractions induced by PP. It should be noted that lactate exhibits protective effects against inflammation in the colon [35]. Therefore, the direct application of PP into the colon may be beneficial to colon physiology.

In these in vitro experiments, PP with neutral pH = 7.4 was used. However, non-neutralized 10% PP has a pH of about 3, and the direct application of this concentration to the colon induces more powerful contractions than those of PP with a neutral pH. Low fecal pH has been shown to stimulate intestinal propulsion.

PP also stimulates local circulation via the activation of cells of the cardiovascular system [6,27]. Thus, the direct application of PP via enema to the colon could suppress pathogenic microbiota and stimulate colon motility and local circulation. It should be noted that the colonic application of PP via enema is one of the suggested routes of PP administration to stimulate mucosa-associated lymphatic tissue [6,27].

## 3. Materials and Methods

### 3.1. Animals

Newborn male Wistar rats (1–2 days old) from the vivarium of our institute were used in the experiment. Newborn rats were euthanized by CO_2_ inhalation in a special plastic chamber.

### 3.2. Rat Colon Isolation and Recording of Contractile Activity

The distal colon (3 mm from the anus) was quickly removed and cleaned at room temperature. Two aluminum clips were individually affixed to opposing ends of the colon segment, which was 5 mm-long and approximately 0.5 mm-wide. One clip was attached to a fixed hook on one side, and the other clip was attached to a tenzo transducer. The sensitivity of the tenzotransducer was 1 μN. Each colon segment was placed horizontally into an experimental chamber containing 2 mL Krebs solution (KS, for composition see below). The temperature of KS in the chamber was maintained at 37 °C, and it was oxygenated with 95% O_2_ and 5% CO_2_. The tenzotransducer was connected to an Endim 620.02 recorder (RFT, Berlin, Germany). An initial tension load (around 0.2–0.4 mN) was applied to the colon tissue, and then the tissue was held for 30 min with fresh KS before PP application. PP was applied in cumulative doses.

### 3.3. Registration of Intracellular Calcium

For calcium registration, the colon was incised longitudinally, and colon tissue (samples were 3 mm-long and approximately 1 mm-wide) was immersed in the KS solution with Fura 2 AM (Sigma-Aldrich, St. Louis, MO, USA) at a concentration of 10 μM for 1 h at 37 °C in a CO_2_ incubator (Jouan, Orléans, France) at 5% CO_2_. Then, the colon tissue was washed with KS solution for 30 min, and intracellular calcium, [Ca^2+^]_i_ was measured. The colon tissue was placed in a chamber (0.5 mL) with a thin glass bottom, and it was fixed along the edges with plexiglass clips at 37 °C and was oxygenated with 95% O_2_ and 5% CO_2_. [Ca^2+^]_i_ was measured using a computerized ratiometric system for an intracellular ion content assay (Intracellular Imaging and Photometry System), employing an inverted Nikon TMS microscope (Tokyo, Japan, objective 30×) equipped with a monochrome digital video camera RS_170 CCD (Cohu, Inc., San Diego, CA, USA) and the associated software. The tissue was illuminated at wavelengths of 340 and 380 nm, and the fluorescence emission was registered at 510 nm. The built-in software InCyt_Im2^TM^ allowed for measuring the intracellular calcium ion concentration as a ratio of the fluorescence emission intensities excited at 340 and 380 nm (*F*340/*F*380), using a calibration curve based on the calibration solutions (Fluka Chemie GmbH, Buchs, Switzerland) with known Ca^2+^ concentrations. Measurements of [Ca^2+^]_i_ and tension registrations were carried out independently on different colon segments.

### 3.4. Drugs and Reagents

This study utilized a proprietary probiotic/pharmabiotic product (PP) produced according to U.S.-patented technology [27]. Various food products with specially selected lactobacilli were fermented for more than three days to achieve maximal bacterial metabolite concentrations [27]. A detailed description of the ingredients, fermentation time and types of lactobacilli are given in the U.S. patent in [27]. The resulting liquid portion (free of bacteria) (PP) was utilized in all experiments; to remove fermenting bacteria, it was centrifuged at 5000× *g* for 15 min at room temperature and was then filtered through Millipore (MilliporeSigma, Burlington, MA, USA) filters (0.22 μm). The pH of PP was around 3 and was neutralized to a pH of 7.4 by the appropriate addition of 1 M NaOH immediately before the experiments in vitro. As the main end product of fermentation, PP contained 60–90 mM lactate.

The Krebs solution (KS) had the following composition: 120 mM NaCl, 5 mM KCl, 1.2 mM NaH_2_PO_4_, 1.2 mM MgCl_2_, 15.4 mM NaHCO_3_, 2.5 mM CaCl_2_ and 11.5 mM glucose with a pH of 7.4. Acetylcholine chloride was purchased from Sigma-Aldrich and was diluted from a 10 mM stock during the experiments.

### 3.5. Statistical Analysis and Ethical Approval

All data give the mean ± SEM; *n* indicates the number of experiments. All animal care and treatment procedures were performed in accordance with the Animal Welfare Act and the Institute Guide for Care and Use of Laboratory Animals.

## 4. Conclusions

Lactobacillus metabolites produced by a special fermentation method quickly stimulate colon contractions and an increase in intracellular calcium. The direct application of PP via enema to the colon could stimulate colon motility and suppress pathogenic microbiota, owing to the antagonistic property of PP towards pathogens.

## 5. U.S. Patent

Sobol CV, Sobol YuTs. Composition and method for producing and the use of a fermented hydrolyzed medium containing microorganisms and products of their metabolism. U.S. Patent 6953574B2, 2005.

## Figures and Tables

**Figure 1 ijms-24-00662-f001:**
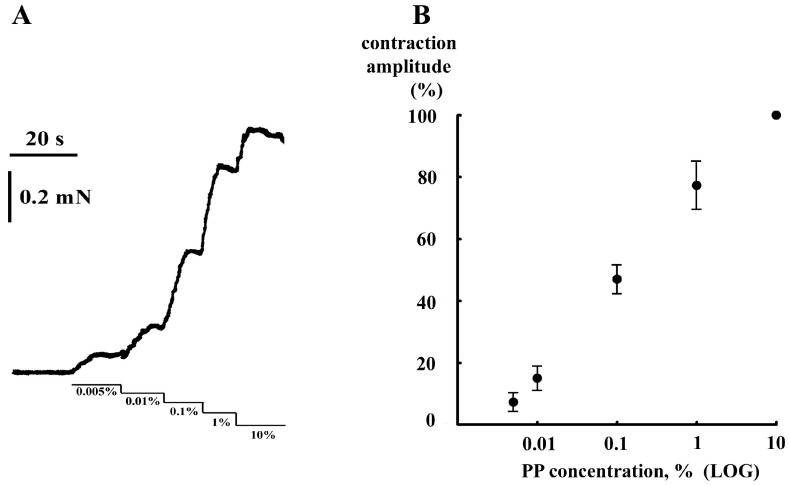
Contractions in rat colon segment in response to cumulative PP application. (**A**) Experimental data of one experiment. The duration and concentrations of PP are shown by horizontal lines under the registration of the contraction. Vertical mark—force of contraction (mN); horizontal—time (s). (**B**) Dose–response curve for four independent experiments. Ordinate—the amplitude of contraction normalized to the contraction induced by 10% PP; abscise—concentration of PP/logarithmic scale (%).

**Figure 2 ijms-24-00662-f002:**
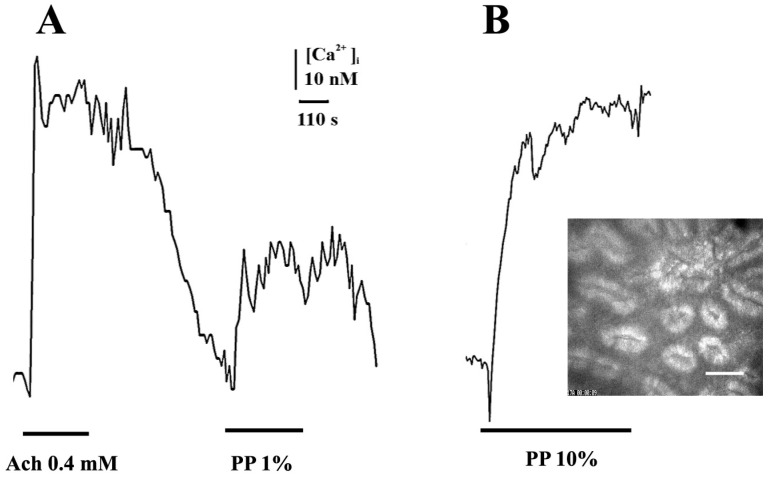
Calcium dynamics in rat colon segment in response to application of various concentrations of PP and acetylcholine (Ach). The duration and concentrations of PP and Ach are shown by horizontal lines under the registration of calcium curves. (**A**) Ach (0.4 mM and 1% PP). (**B**) 10% PP. Vertical mark—concentration of intracellular calcium (nM); horizontal—time (s). The colon segment is shown in the insert, the fluorescence is at 340 nM plus visible light and the mark is 50 μm.

## Data Availability

Intracellular Imaging and Photometry System has been described online https://www.intracellular.com/.

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
