# Peer review of "Stimulatory Effect of Lactobacillus Metabolites on Colonic Contractions in Newborn Rats"

_ijms, 2022, doi:10.3390/ijms24010662_

Round 1
Reviewer 1 Report
Dear Authors:
Thank you for your submission.
I read your manuscript to find something wrong.
As the following files:
Thank you.

Author Response
Dear Reviewer,
I thank you for your review and valuable comments.
Below are detailed responses to your comments. English language has been edited by MDPI Publisher.
Your manuscript is a brief report. Your manuscript title is “Stimulatory Effect of Lactobacillus Metabolites on Colonic Contractions in Newborn Rats”. Your manuscript has 9.5% PlagScan rang. I think if you can lower this range will be better. I hope it may be at 5%. This is only my recommendation. I think that’s good.
As the following recommendation:
I would be very grateful if you could send the text with marked plagiarisms so that I can correct the sensitive places.
- Materials and methods ……..then Results…….then…..Discussion…Conclusions……
It's corrected according to your recommendation.
- Abstract
Need to write more, too short. Almost 250 words.
Need to write background, methods, results, and conclusion.
Abstract is rewritten as structured abstract as recommended. Graphical abstract is added.
- Introduction
Combination all paragraphs to become one.
Cities [5, 6, 8, 9] to [5, 6-9]
It's corrected according to your recommendation
- Material
Too complex. Need 3.1….3.2….3.3. too short. Need more words.
Material and methods are rewritten as recommended
- Conclusion needs to write more.
Conclusion is extended
- References need to rewrite it.
I have added three references (5,10,23) published recently and related to the theme of the article and deleted two my references.
- Results:
equal to 0.13±0.02% (n=4) From where, ….
I have added the dose–response curve for four independent experiments to Figure 1 as Figure 1B and described the maximum dose as 10% of PP. The description of ED50 was added in the text.

Reviewer 2 Report
A very interesting topic, with clinical potential. All phases of the research were adequately conducted and described in the paper.
line 13: Please describe the "US"abbreviation
line 36: agerelated : Age -related
English language needs polishing.
Author Response
Dear Reviewer,
I thank you for your review and valuable comments.
English language has been edited by MDPI Publisher.
Best regards,
Constantin Sobol

Reviewer 3 Report
General comments
This is an interesting topic but it is a single author paper which raises some concerns as there is lack of a group to over see the data and no way of avoiding bias nor guarantee of authentic data
Given that the author holds a patent on the product used there is plainly a conflict of interest. Really we need to see if a researcher who does not have CoIs can reproduce
Specific comments
1. Results
The results are shown by means of 2 images of single experiments. There is no way of knowing how reproducible such effects or not whether these are simply the most dramatic.
2. Interpretation
Translating such data to the in vivo situation is very difficult if not impossible since it is quite unclear what the relevant concentrations are at the level of colonic smooth muscle given the mucus, epithelium and submucosal structure which are between the gut lumen and the smooth muscle.
3. The discussion goes far beyond the experimental data presented . It should focus on the experiment and its weakness together with data on the concentrations to be found in the gut.
Author Response
Dear Reviewer,
I thank you for your review and valuable comments.
Below are detailed responses to your comments. English language has been edited by MDPI Publisher.
General comments
This is an interesting topic but it is a single author paper which raises some concerns as there is lack of a group to over see the data and no way of avoiding bias nor guarantee of authentic data
Given that the author holds a patent on the product used there is plainly a conflict of interest. Really we need to see if a researcher who does not have CoIs can reproduce
In fact, the ingredients and production are described in details both in the patent and in the public press.
Sobol C.V. A new class of pharmabiotics with unique properties. In: Soft Chemistry and Food Fermentation, Grumezescu, A.M., Holban A.M., Eds.; Elsevier: Amsterdam, The Netherlands, 2017, Volume 3, pp. 79–112.
Therefore, similar products and effects are reproducible.
For example, Dr Neto D.P. with colleagues reproduced my experimental protocol on enteric nerve cells using Akkermansia muciniphila supernatants (2021).
https://doi.org/10.1016/j.isci.2022.103908
In his thesis he referred to my earliest work.
In other study it was demonstrated that cell-free supernatants produced by Lactobacillus strains revealed antibacterial activity in vitro.
https://doi.org/10.3389/fmicb.2019.01403
These data are comparable with the results that I have published earlier.
Specific comments
- Results
The results are shown by means of 2 images of single experiments. There is no way of knowing how reproducible such effects or not whether these are simply the most dramatic.
I have added the dose–response curve for four independent experiments to the Figure 1 as Figure 1B and described the maximum dose as 10% of PP. The description of ED50 was added in the text.
These effects can be reproduced.
- Interpretation
Translating such data to the in vivo situation is very difficult if not impossible since it is quite unclear what the relevant concentrations are at the level of colonic smooth muscle given the mucus, epithelium and submucosal structure which are between the gut lumen and the smooth muscle.
- The discussion goes far beyond the experimental data presented. It should focus on the experiment and its weakness together with data on the concentrations to be found in the gut.
I agree that the concentration of the product cannot be reliably determined in the colon. Moreover, the dose-effect may differ from in vitro experiments. However, the main property of the product to cause defecation, especially with constipation, is clearly expressed. I was convinced of this from my own experience and in the clinic in isolated cases. Therefore, in experiments in vitro, I investigate and discuss precisely these effects.
I have added three references (5,10,23) published recently and related to the theme of the manuscript and deleted two my references.

Round 2
Reviewer 1 Report
Dear Author:
Thank you for revising this manuscript.
I can agree to accept it.